# Scaling Hierarchical Coreference
# with Homomorphic Compression

**Michael Wick**                                    MICHAEL.WICK@ORACLE.COM
*Oracle Labs,*
*Burlington, MA*

**Swetasudha Panda**                          SWETASUDHA.PANDA@ORACLE.COM
*Orable Labs,*
*Burlington, MA*

**Joseph Tassarotti**                                    JTASSARO@MIT.EDU
*Massachusetts Institute of Technology*
*Cambridge, MA*

**Jean-Baptiste Tristan**                  JEAN.BAPTISTE.TRISTAN@ORACLE.COM
*Orable Labs,*
*Burlington, MA*

## Abstract

Locality sensitive hashing schemes such as simhash provide compact representations of multisets from which similarity can be estimated. However, in certain applications, we need to estimate the similarity of dynamically changing sets. In this case, we need the representation to be a homomorphism so that the hash of unions and differences of sets can be computed directly from the hashes of operands. We propose two representations that have this property for cosine similarity (an extension of simhash and angle-preserving random projections), and make substantial progress on a third representation for Jaccard similarity (an extension of minhash). We employ these hashes to compress the sufficient statistics of a conditional random field (CRF) coreference model and study how this compression affects our ability to compute similarities as entities are split and merged during inference. We also provide novel statistical analysis of simhash to help justify it as an estimator inside a CRF, showing that the bias and variance reduce quickly with the number of hash bits. On a problem of author coreference, we find that our simhash scheme allows scaling the hierarchical coreference algorithm by an order of magnitude without degrading its statistical performance or the model's coreference accuracy, as long as we employ at least 128 or 256 bit hashes. Angle-preserving random projections further improve the coreference quality, potentially allowing even fewer dimensions to be used.

## 1. Introduction

Probabilistic models in machine learning, such as conditional random fields (CRFs), are widely successful at modeling many problems at the heart of knowledge base construction, including those in natural language processing, information extraction and data integration. However, when dealing with natural language data, the underlying feature representations are often sparse, high-dimensional and dynamic (*i.e.,* they change during inference). In this paper we consider the task of coreference resolution, in which the goal is to partition a set of mentions into the entities to which they refer. We might represent each mention with a

feature vector in which each dimension corresponds to a word or $n$-gram. Since only a small subset of the vocabulary is observed per mention, most elements of the vector are zero.

Given the model and these representations, inference entails making decisions about whether two entities should be coreferent. To make such decisions, the model estimates a probability that involves computing the similarities between the aggregate feature representations of the two entities' mentions. Since *all* the vectors are both sparse and high-dimensional, these similarity operations are computationally expensive because the sparse data structures supplant the dense arrays that would otherwise support fast look-ups. Moreover, as the inference algorithm makes decisions about whether or not two entities are coreferent, we may have to split or merge the entities and thus we must *update the feature vector* to reflect these changes. Maintaining such sparse-vector representations in the inner-loop of probabilistic inference is expensive, especially as the entities grow in size.

In order to cope with the computational problems associated with sparse, high dimensional dynamic feature representations, we propose using *homomorphic compression*, in which the compressed representations of intermediate inference results can be computed directly from their operands, allowing inference to run directly on the compressed representations of the data even as they change. In this paper, we consider several such schemes to scale hierarchical coreference. First, we propose a novel homomorphic cosine-preserving hashing scheme based on simhash [Charikar, 2002] that also supports addition and subtraction to more efficiently represent the data and the evolving intermediate results of probabilistic inference. Second, because various linear norm-preserving random projections also preserve angles [Magen, 2002], we can directly compute cosine similarity on projected data – linearity of the projections means that they too can be updated dynamically. The resulting angle estimates are superior to the homomorphic simhash representation, at the cost of reduced efficiency for certain operations. Third, we develop a homomorphic version of minhash [Broder, 1997b] to support Jaccard similarity. Our current algorithm is biased, but the bias appears small in practice for the situations we have considered. Although the minhash based set representations are not currently employed in hierarchical coreference, they might be useful in other models or applications that involve binary features over changing sets [Culotta et al., 2007b].

We provide error analysis for all three schemes, collating and extending known results, and in the case of simhash, we provide novel statistical analysis for its use as a direct estimator for $\cos(\theta)$ that shows the bias and variance decrease rapidly with the number of bits, helping to justify its use in a CRF for a task like coreference. On a hierarchical model for coreference resolution, the proposed simhash scheme improves the speed of probabilistic inference by an order of magnitude while having little effect on model quality. Moreover, we find that the underlying random projection representation can provide even better cosine estimates than simhash, at the cost of not being able to use certain fast bitwise-operations to compute similarities. Finally, we briefly evaluate homomorphic minhash and show that even though there are possible pathological cases, as long as we employ enough hash functions, the estimates are reasonably close to the true Jaccard, albeit, biased.

## 2. Coreference Resolution: Background and Challenges at Scale

**Coreference resolution**    Coreference resolution is the problem of determining whether different *mentions* refer to the same underlying *entity* [Getoor and Machanavajjhala, 2012].

For example, in the sentence "In a few years [McDaniels] would replace [Belichick] as the forty-two year old [quarterback], [Tom Brady] retires." coreference must correctly determine that "quarterback" refers to Tom Brady and not Belichick or McDaniels. This type of coreference is sometimes referred to as noun-phrase or within document coreference and often relies upon linguistic and discourse motivated features [Raghunathan et al., 2010]. Coreference resolution arises in many other situations; for example, when merging two or more databases together it is desirable to remove duplicates that result from the merge, a problem sometimes termed record linkage or deduplication [Newcombe et al., 1959]. Coreference is also foundational to *knowledge base construction* which requires that we combine information about entities of interest from multiple sources that might mention them in different contexts. For example, if we were to build a knowledge base of all scientists in the world — similar to Google scholar — we would need to perform the task of *author coreference* to determine who authored what papers [Han et al., 2004, Culotta et al., 2007a]. Are the following two mentions of "J Smith" the same author?

> *V Khachatryan, AM Sirunyan,...,* **J Smith.** *Observation of the diphoton decay of the Higgs boson and measurement of its properties. The European Physical Journal 2014*

> *S Chatrchyan, V Khachatryan, AM Sirunyan, A Tumasyan, W Adam,* **J Smith**. *Jet production rates in association with W and Z bosons in pp collisions. J High Energy Physics 2012*

Although generally this is a difficult problem, it can be solved with machine learning since features of the mentions such as the words in the title (both have "Boson" in common), the topic of the title (both are about a similar subfield of physics), the journal (both are physics journals) and the co-authors (there appears to be a co-author in common) provide some evidence about whether or not the two "J Smith's" might be the same person.

In order to solve the problem, it is thus common to extract such contextual features about each mention, such as in the above example, features from the title, co-author list, venue, year and author-name and employ them in a probabilistic model. These features are typically the raw words, character-ngrams and normalized variants thereof, sometimes with positive real-valued weights to indicate the importance (e.g., via TFIDF) of each feature. Then, given such features, a coreference model measures the similarities between mentions via functions such as cosine-similarity. In contrast to within document coreference discussed earlier, this type of coreference resolution problem is better suited for similarity based models, such as the ones we will use in the following. Moreover, since coreference decisions are not restricted by document-boundaries, the entities can grow unbounded in size, making compact representations of their growing feature sets especially important.

Typically, the model is a discriminative conditional random field (CRF) that measure the probability of an assignment of mentions to entities conditioned on the observed features [McCallum and Wellner, 2003]. The model factorizes into potentials that score local coreference decisions. Local search procedures such as greedy-agglomerative clustering or Markov-chain Monte Carlo (MCMC) find the most likely assignment of mentions to entities [McCallum and Wellner, 2003, Culotta et al., 2007a, Wick et al., 2012].

In *pairwise* models, potential functions measure the compatibility of two mentions being in the same cluster. An issue with such models is that the possible pairwise comparisons scales

quadratically with the number of mentions. An alternative class of models that avoids this quadratic blow-up are *entity-based*, in which entities are treated as first-class variables with their own set of inferred features, and potentials measure compatibility between mentions and entities. However, entity-based models come with their own scalability challenges. To illustrate the problem (and our solution), we focus on an entity-based model called *hierarchical coreference*, which recently won a challenge to disambiguate inventor names for the USPTO, due to its accuracy and scalability [Uni, 2016, Monath and McCallum, 2016].

**Hierarchical Coreference**    In the hierarchical coreference model, mentions are organized into latent tree structures [Wick et al., 2012]. There is one tree per entity with mentions at the leaves and intermediate nodes as "subentities" that organize subsets of the entity's mentions. Rather than modeling interactions between mention-pairs, the potential functions measure compatibility between child and parent nodes in the tree. The score of a given assignment of mentions into latent trees is the product of all model potentials which includes these child-parent compatibility scores as well as some additional priors on tree-shape and entities. These compatibility scores are parametrized cosine functions.

Each mention is endowed with multiple feature variables that each capture a subset of the total features. For example, in author coreference, one feature variable might capture features of the author's name and another might capture features of the title and venue of the paper. Colloquially, we refer to these feature variables as "bags" since they inherit the usual bags-of-words assumption. We distinguish between different "bag-types" which each capture different types of features (e.g., the author name bag, title words bag, co-author bag, etc). The other nodes in the tree also contain feature variables (bags), but the values of these variables are determined by the current assignment of children to that parent node. In particular, *a parent's bag is the sum of all its children's bags.* In order to maintain this invariant, the bag representations must be updated to reflect the current state of inference, and hence for efficiency reasons, representations must be homomorphic with respect to operations that will be performed on the bags.

Interpreting bags as vectors, the cosine distance between them is used to calculate their compatibility. The primary potential functions measure the compatibility between each child's bag and its parent's bag. There is one potential for each bag-type. For example, to measure the compatibility between a node $z_i$ and $z_j$, let $y_{ij}$ be the binary variable that is 1 if and only if $z_j$ is the parent of $z_i$, and let $b_i^{(0)}$ and $b_j^{(0)}$ be a bag for $z_i$ and $z_j$ respectively, then the potential $\psi^{(0)}$ for "bag 0" scores a coreference decision as:

$$\psi^{(0)}(z_i, z_j, y_{ij}; w, t) = \begin{cases} 1 & y_{ij} = 0 \\ \exp\left(w(\cos(b_i^{(0)}, b_j^{(0)} - b_i^{(0)}) - t)\right) & y_{ij} = 1 \end{cases} \tag{1}$$

where $w$ is a real-valued weight and $t$ is a real-valued translation parameter for potential $\psi^{(0)}$. The potentials for each bag-type have parameters $w, t$ that we can fit to data.

Because only a small handful of features are ever observed for a given mention, typical implementations of hierarchical coreference employ sparse-vector representations to represent bags (*e.g.,* the implementation found in FACTORIE [McCallum et al., 2008, 2009]).

However, a key disadvantage of sparse vector representations is that they must store the indices and weights of the non-zero elements, which means that the data-structures

must dynamically change in size as MCMC splits and merges entities. As the sizes of the entities grow, these operations become increasingly expensive. Similar issues arise in other entity-based models where features of entities are aggregated from those of mentions. Thus, while entity-based models avoid the quadratic comparison issue of pairwise models, a straight-forward sparse representation of their feature vectors is no longer efficient.

Is there an alternative representation of feature vectors which (a) allows fast evaluation of cosine-similarity, and (b) can be efficiently dynamically updated? As we describe in the next section, the simhash hashing function [Charikar, 2002] provides a representation with property (a). However, the standard simhash cannot be updated as vectors are modified. We therefore develop a variant which we call *homomorphic* simhash, which has both properties (a) and (b). We also identify two other schemes that support these properties.

## 3. Homomorphic Representations for Measuring Similarity

We now discuss three different homomorphic representations that support addition and subtraction in the compressed space, while preserving similarity estimates. We propose homomorphic simhash and a related random projection for cosine similarity of high-dimensional vectors and multisets; and homomorphic minhash for Jacard similarity of sets.

### 3.1 Homomorphic simhash and its Statistical Properties

**Background: simhash**    A *locality-sensitive hash function* for a distance metric $d$ on a set of objects $S$ is a function $H$ such that given $x, y \in S$, we can estimate $d(x, y)$ from the hashed representations $H(x)$ and $H(y)$. Simhash [Charikar, 2002] is a locality sensitive hash function for cosine similarity.

To understand simhash, it is first helpful to consider the following randomized process: Imagine that we have two vectors $a$ and $b$ on the unit hypersphere in the Euclidean space $\mathcal{R}^d$ with angle $\theta$ between them, and we want to produce an estimate of $\cos(\theta)$. Select a random $d$-dimensional vector $u$ by sampling each of its coordinates independently from $N(0, 1)$. Let the random variable $X$ have value 1 if $a$ and $b$ are on different sides of the hyperplane orthogonal to $u$, and 0 otherwise. Then $X$ is a Bernoulli random variable with expectation:

$$\mathbb{E}[X] = 1 - \mathbb{P}(sign(a \cdot u) = sign(b \cdot u)) \tag{2}$$

$$= 1 - \frac{\theta}{\pi} \tag{3}$$

Let $X_1, ..., X_n$ be the result of independently repeating this process several times, and set $\overline{X}_n = \frac{1}{n} \sum_{i=1}^{n} X_i$. Then $\mathbb{E}[\overline{X}_n] = 1 - \frac{\theta}{\pi}$, and hence $\mathbb{E}[\pi(1 - \overline{X}_n)] = \theta$, so that we can use[1] $\cos(\pi(1 - \overline{X}_n))$ as an estimator of $\cos(\theta)$.

The idea behind simhash is to come up with a hash representation which lets us reproduce this randomized estimator: to construct a function $H$ which produces $n$-bit hashes, we first randomly sample $n$ vectors $u_1, \ldots, u_n$ as above. Then, given a vector $a$, the hash $H(a)$ is the length $n$ bit sequence in which the $i$th bit is 1 if the sign of $a \cdot u_i$ is positive and 0

---

1. Charikar [Charikar, 2002] notes that for some applications, $1 - \frac{\theta}{\pi}$ may be a good enough approximation of $\cos(\theta)$, so that one can use $\overline{X}_n$ directly as an estimator of $\cos(\theta)$.

otherwise. Now, from two hashed representations $H(a)$ and $H(b)$, if the $i$th bit in the two hashes disagree, this is equivalent to $X_i$ in the randomized process above being equal to 1. Thus, by counting the number of positions where the hashes are distinct and dividing by $n$, we get $\overline{X}_n$, thereby producing an estimate of $\cos(\theta)$.

Rather than constructing the hash function $H$ by sampling the $u_1, \ldots, u_n$ vectors from the $d$-dimensional unit sphere uniformly, a common optimization is to instead sample them from $\{-1, 1\}^d$. This has two advantages. First, it is faster to compute the dot product since no floating point multiplication is involved. Second, rather than having to explicitly sample and store each $u_i$ as a vector, we can replace each $u_i$ by a 1-bit feature hash function $h_i$: the "value" of the vector represented by $h_i$ is 1 at coordinate $j$ if $h_i(j) = 1$ and is $-1$ if $h_i(j) = 0$. We write $a \cdot h_i$ for the dot product of $a$ with the vector corresponding to $h_i$.

By restricting only to test vectors with coordinates of the form 1 and $-1$, the corresponding expected value of $\pi(1 - \overline{X}_n)$ is no longer exactly $\theta$ (see [Leskovec et al., 2014, Section 3.7.3] for an example), but for high-dimensional spaces, this approximation is known to be effective in practice [Henzinger, 2006].

**Homomorphic simhash**   If we want to use simhash as a representation of feature vectors for entities in coreference resolution, then we need to be able to update the simhash representation as entities are merged and split. In particular, if we join nodes with feature vectors $a$ and $b$, then the vector of their new parent will be $a + b$. However, if we only store $H(a)$ and $H(b)$, rather than the vectors $a$ and $b$ themselves, we cannot compute $H(a + b)$: the $i$th bit of $H(a)$ and $H(b)$ just records the sign of $a \cdot h_i$ and $b \cdot h_i$, and if these are different, we do not know what the sign of $(a + b) \cdot h_i$ should be. A similar problem occurs when we split a child with vector $b$ from a parent with vector $a$, since the updated parent's hash should be $H(a - b)$.

Our solution is instead to store the actual dot product of $a \cdot h_i$ in the hash of $a$, rather than just the sign. That is, $H(a)$ is now an array of length $n$ instead of an $n$-bit sequence. And since

$$(a + b) \cdot h_i = a \cdot h_i + b \cdot h_i \qquad \text{and} \qquad (a - b) \cdot h_i = a \cdot h_i - b \cdot h_i$$

we can compute $H(a + b)$ by adding component-wise the arrays for $H(a)$ and $H(b)$, and similarly for $H(a - b)$. Finally, we can still efficiently compute the cosine distance between two vectors $a$ and $b$ by examining the signs of the entries of $H(a)$ and $H(b)$. We call this representation *homomorphic* because $H$ is a homomorphism with respect to the additive group structure on vectors.

Of course, storing each dot product instead of just the signs increases the size of our hashes. However, they are still small compared to the feature vectors and, more importantly, their sizes are fixed. We can also store both the dot product and the signs as a bit vector, making sure to update the sign vector after each operation based on the dot product. By storing the sign vector separately we can quickly count the signs in common between two vectors using bitwise operations.

**Statistical Properties**   Recall that since $\mathbb{E}[\pi(1 - \overline{X}_n)] = \theta$, we can derive a plausible estimate of $\cos(\theta)$ from $\overline{X}_n$. In particular, let $g(x) = \cos(\pi(1 - x))$ and consider the estimator $C_n = g(\overline{X}_n)$. We now describe some statistical properties of $C_n$. Our emphasis here is somewhat different from the standard analyses found in related work. The reason is that

LSHs like simhash are most commonly used for duplicate detection [Henzinger, 2006] and for approximate nearest neighbor search [Leskovec et al., 2014], which is quite different from our use case. In those settings, one wants to show that if two items $x$ and $y$ are very similar, then the distance estimated from $h(x)$ and $h(y)$ will very likely be quite small, and conversely if $x$ and $y$ are very different, then their estimated distances will be large. In such cases, the linear approximation to cosine $\overline{X}_n$ is sufficient. However, since we want to use the cosine distance estimates $C_n$ as part of the potential function in our CRF, we are interested in additional statistical properties of the estimator.

**Lemma 3.1.** $C_n$ is consistent. In particular, $C_n \xrightarrow{a.s.} \cos(\theta)$.

*Proof.* By the strong law of large numbers, we have that $\overline{X}_n \xrightarrow{a.s.} 1 - \frac{\theta}{\pi}$. Since $g$ is continuous, by the continuous mapping theorem [Van Der Vaart, 1998],

$$g(\overline{X}_n) \xrightarrow{a.s.} g(1 - \frac{\theta}{\pi}) = \cos(\theta)$$

□

**Lemma 3.2.** $\mathbb{E}[C_n] = \cos(\theta) + E_n$, where $|E_n| \leq \frac{\pi^2}{8n}$

*Proof Sketch.* Set $\mu = \mathbb{E}[\overline{X}_n] = 1 - \frac{\theta}{\pi}$. The first degree Taylor series for $g(x)$ about $\mu$ is:

$$g(x) = \cos(\theta) + \pi \sin(\theta)(x - \mu) + R(x)$$

where $R$ is the remainder term. We have then that:

$$\mathbb{E}[g(\overline{X}_n)] = \cos(\theta) + \pi \sin(\theta)(\mathbb{E}[\overline{X}_n] - \mu) + \mathbb{E}[R(\overline{X}_n)] \tag{4}$$
$$= \cos(\theta) + \mathbb{E}[R(\overline{X}_n)] \tag{5}$$

Thus it suffices to bound $|\mathbb{E}[R(\overline{X}_n)]|$, which can be done using Lagrange's remainder formula (see appendix). □

**Lemma 3.3.** $\mathbb{V}[C_n] = \frac{\pi^2 \sin^2(\theta)}{n} \cdot \frac{\theta}{\pi}(1 - \frac{\theta}{\pi}) + O(n^{-3/2})$

*Proof Sketch.* For intuition, note that the Taylor series above for $g$ shows us that $g(x) \approx \cos(\theta) + \pi \sin(\theta)(x - \mu)$. So, recalling that $\mathbb{V}[C_n] = \mathbb{V}[g(\overline{X}_n)] = \mathbb{E}[g(\overline{X}_n)^2] - \mathbb{E}[g(\overline{X}_n)]^2$, and plugging in the approximation we get:

$$\mathbb{V}[g(\overline{X}_n)] \approx \mathbb{E}[(\cos(\theta) + \pi \sin(\theta)(\overline{X}_n - \mu))^2] - \cos(\theta)^2 \tag{6}$$
$$= 2\pi \sin(\theta)\mathbb{E}[\overline{X}_n - \mu] + \pi^2 \sin^2(\theta)\mathbb{E}[(\overline{X}_n - \mu)^2] \tag{7}$$
$$= \pi^2 \sin^2(\theta)\mathbb{E}[(\overline{X}_n - \mu)^2] \tag{8}$$

To obtain the actual error bound, we carry out the same process but without dropping $R(x)$ from the Taylor approximation for $g$, and then once again use Lagrange's remainder formula to bound the remainder (see appendix). □

Finally, since $C_n$ is equal to a Lipschitz continuous function of $n$ independent Bernoulli random variables, we can use the the method of bounded differences [Dubhashi and Panconesi, 2009, Corollary 5.2], to derive the following Chernoff-like tail bound:

**Lemma 3.4.** $\mathbb{P}(|C_n - \cos(\theta)| > \delta + \frac{\pi^2}{8n}) \leq 2e^{-2n\delta^2/\pi^2}$.

## 3.2 Angle Preseving Random Projections for Cosine

The statistics underlying simhash, whether the hyperplanes are drawn from Gaussians or the $d$-dimensional hypercube (i.e., Rademacher distributions), are actually random projections for which the Johnson-Lidenstrauss lemma applies [Johnson and Lindenstrauss, 1982, Indyk and Motwani, 1998, Achlioptas, 2003]. The lemma states that any set of $m$ points in $d$-dimensional Euclidean space can be embedded into $n$-dimensional Euclidean space, where $n$ is logarithmic in $m$ and independent of $d, n = O(\epsilon^{-2} \log m)$, such that all pairwise distances are maintained within an arbitrarily small factor $1 \pm \epsilon$, where $0 < \epsilon < 1$. Since the projections are linear, they are homomorphic with respect to addition and subtraction. Moreover, previous work shows that the norm-preserving property of a linear random projection $A$ implies an angle-preserving property [Magen, 2002]; that is, for vectors $u, v$, let $\theta = \sphericalangle(v, u)$ and $\hat{\theta} = \sphericalangle(Av, Au)$. The result is the following:

$$\theta - \hat{\theta} \le 2\sqrt{\epsilon} \text{ and } \frac{1}{1 + \epsilon'} \le \hat{\theta}/\theta \le 1 + \epsilon' \tag{9}$$

where $\epsilon \le \frac{1}{3}, \epsilon' = \frac{8}{\pi}\sqrt{\epsilon}$ and $n \ge 60\epsilon^{-2} \log m$. Therefore, we are justified in using the statistics underlying simhash to directly estimate the cosine similarity; viz., $\cos\theta \approx \cos\hat{\theta}$. More precisely, using a Taylor expansion, $|\cos\theta - \cos\hat{\theta}| \le |2\sqrt{\epsilon}\sin\hat{\theta} + \frac{(2\sqrt{\epsilon})^2}{2}\cos\hat{\theta}| \le 2\sqrt{\epsilon(1 + \epsilon)}$. Although we lose the bit-representation that supports the fast bit-operations that makes simhash so efficient in applications such as webpage deduplication that employ the linear estimator, we potentially gain an improvement in the quality of the cosine estimate. Certainly, the estimate will be smoother since simhash essentially quantizes the angles into a finite number of possibilities equal to the number of bits. Since the same representation supports both types of estimates, we could even alternate between the estimation strategies as dictated by the particular situation: if a large number of similarity comparisons are performed for each entity then simhash makes more sense, otherwise a direct computation of cosine makes more sense. Regardless, as we will later see, both schemes provide sufficiently fast and accurate cosine estimates for coreference resolution.

## 3.3 Homomorphic Minhash for Jaccard Similarity

Minhash [Broder, 1997a, Broder et al., 2000] is a locality-sensitive hash function for Jaccard similarity, defined for binary vectors (encoding sets). The Jaccard similarity between two sets $S_1, S_2 \in \Omega$ is $J = \frac{|S_1 \cap S_2|}{|S_1 \cup S_2|}$. Minhash applies a random permutation (which in practice is accomplished using a random hash function, such as seeded 64-bit murmur hash) $\pi : \Omega \to \Omega$, on a given set $S \subset \Omega$ and then extracts the minimum value $h_\pi(S) = \min(\pi(S))$. The probability that the minhash function for a random permutation of a set computes the same value for two sets is equal to the Jaccard similarity of the two sets [Rajaraman and Ullman, 2010]. In practice multiple ($n$) hash functions are employed to reduce the variance, resulting in a vector $v^S = \langle h_{\pi_0}(S), \cdots, h_{\pi_{n-1}}(S) \rangle$ of $n$ minimum values, one per hash function.

There are three methods we need to design for homomorphic minhash: `union`, `difference` and `score` (to compute the Jaccard estimate). However, unlike simhash for cosine similarity, the operations underlying the statistics for minhash are non-linear due to the elementwise minimum operation that produces the underlying vector of hash values. Moreover, the set semantics on which the minhash `score` method relies are problematic because the `union`

and `difference` methods need to maintain the invariance that a parent set is equal to the union of a collection of child sets: when we perform a difference operation between a parent set and a child set we cannot simply remove all of the child's elements from the parent since a sibling might also (redundantly) contribute some of those elements to the parent. We sketch a solution to these problems and include details in Appendix E. However, we note that our solution is not complete because there are several corner cases we do not yet handle. Nevertheless, we find that the representation works well in practice.

First, we handle the problem of set-semantics by augmenting the $n$-dimensional minhash representation $v^S$ of each set $S$ with an $n$-dimensional vector of counts $c^S$, in which each dimension corresponds to a minimum hash value (essentially embracing multiset semantics, but in the hash space). The count indicates the number of child sets that contribute the associated hash value. For the union of two sets $S = S_1 \cup S_2$, we either keep the count associated with the smaller hash value if they are different (that is, since we set $v^S = v^{S^\star}$ we set $c_i^S = c_i^{S^\star}$ where $S^\star = \operatorname{argmin}_{S_j \in \{S_1, S_2\}}(v_i^{S_j})$), or sum the counts if they are the same (that is, $c_i^S = c_i^{S_1} + c_i^{S_2}$). For difference we employ the same strategy except we subtract the counts instead of add them. The counts are appropriately ignored when estimating the Jaccard since we want the estimate to reflect sets rather than multisets.

Second, we must also address the fact that the minimum is a non-linear operator. The problem arises when the count associated with a hash value becomes zero. Then, how do we recompute what the new minimum value should be? Recomputing the minimum from scratch is a nonstarter because it would require keeping around the original sparse vector representations that we are trying to supplant. Rewriting the difference in terms of unions is also computationally expensive since it would require traversing the entire tree to check what the new minimum value should be. Instead, noting that minhash typically has hundreds of hash functions (e.g., $n = 256$) for each set, we propose to ignore the hash values associated with zero counts, and employ the remaining hashes to compute the Jaccard. This strategy has consequences for both bias and variance. First, since fewer hashes are employed, the variance increases, and if left unchecked may culminate in a worst case in which all counts are zero and Jaccard can no longer be estimated. However, we can periodically refresh the counts by traversing the trees and hopefully we do not have to do such refresh operations too often in practice. Second, since the hashes associated with zero counts are correlated, the estimate is no longer unbiased. Therefore, as described in Appendix E, we modify the Jaccard estimate to make better use of the zero-counts rather than simply ignore them, and this eliminates the bias in some cases. However, we do not have a solution that works in every case.

Finally, there is also the question of how to perform union and difference for cases in which the count is zero. For now, we ignore the zero counts during these operations, but are currently exploring strategies for incorporating them to further reduce bias and variance.

### 3.4 Additional Compression Schemes

Hierarchical coref involves other features, such as entropy and complexity-based penalties on the bag of words representations of context and topics associated with the entities. Although not a focus of this current study, we note that some of these representations depend on the ratios of $p$-norms that can be estimated with Johnson-Lindenstrauss style representations.

Moreover, there exist hashing schemes to enable fast estimation of entropy. We save the homomorphic versions of these hashes for future work.

## 4. Experiments

In this section we empirically study the two cosine-preserving homomorphic compression schemes, simhash and its related random projection, on a conditional random field (CRF) model of author coreference resolution, the problem introduced in Section 2. First we study simhash and we hypothesize that the this representation will substantially improve the speed of inference for the reasons outlined earlier, but it is less clear how the simhash representation will affect the quality of the model. On the one hand, our theoretical analysis shows that the variance and bias reduce rapidly with the number of bits, but on the other hand, the sheer number of vector comparisons that take place during inference is substantial, making it likely for errors to occur anyway. With this in mind, we study how simhash affects the model quality in our first set of experiments. In our second set of experiments, we study the extent to which it improves inference speed. In a third set of experiments, we compare simhash with the random projection method. Finally, we present initial results for homomorphic minhash to empirically assess how it compares to exact Jaccard, which is important given the increased bias and variance of our method.

**Data**   We employ the REXA author coreference dataset,[2] which comprises 1404 author mentions of 289 authors with ambiguous first-initial last-name combinations: *D. Allen*, *A. Blum*, *S. Jones*, *H Robinson*, *S. Young*, *L. Lee*, *J. McGuire*, *A. Moore*. We split the data such that training set contains mentions of the first five ambiguous names (950 mentions) while the testing set comprises the remaining three ambiguous names (454 mentions). The dataset is highly ambiguous since there are multiple entities with each name. In addition, we also employ the DBLP dataset which contains over one million paper citations from which we extract about five million unlabeled author mentions [Ley, 2002].

**Model**   We investigate homomorphic simhash in the context of the hierarchical coreference model presented in Section 2. We employ two types of feature variables, a "name bag" that represents the features of the author's name and a "context bag" that represents the remaining features in the citation from which the author mention is extracted (co-authors, title, venue, topics, keywords). For more details about the features please see Appendix B.

We employ the implementation of hierarchical coreference available in the FACTORIE toolkit, using FACTORIE's implementation of the variables, the model and the inference algorithm [McCallum et al., 2008]. We additionally implement the simhash variables and potential functions inside this framework. We employ FACTORIE's default inference algorithm for hierarchical coreference which is essentially a greedy variant of multi-try Metropolis-Hastings in which the proposals make modifications to the sub-trees (e.g., move a subtree from one entity to another, or merge two trees under a common root node). More details are in previous work, and the implementation is publicly available in FACTORIE [McCallum et al., 2009, Wick et al., 2012]. We estimate the parameters with hyper-parameter search on the training-set (Appendix B).

---

2. <http://www.iesl.cs.umass.edu/datasets.html>

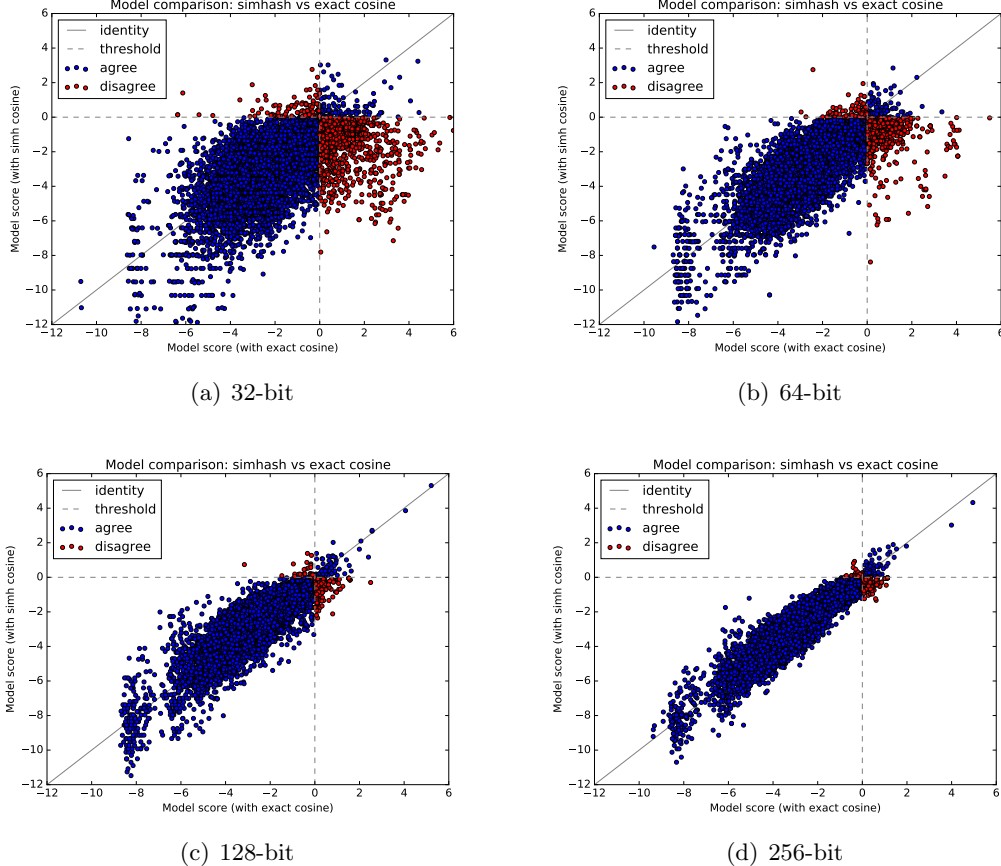

(a) 32-bit

(b) 64-bit

(c) 128-bit

(d) 256-bit

Figure 1: Model score comparison with homomorphic simhash and exact sparse-vector representations. The closer points are to the identity reference line, the better. The dotted horizontal and vertical lines represent the decision threshold for inference. Points for which the two models agree are in blue, the agreements rates are: 88.6, 83.6, 97.0, 97.8 percent (respectively 32, 64, 128, 256 bits).

**Experiment 1: Simhash Estimation Quality**  Here we study the quality of models with simhash representations by comparing them to models with exact representations. First, we compare how these models evaluate and score the intermediate results of MCMC inference. We run MCMC for 100,000 steps to optimize simhash-based models on the REXA test set (with either 256, 128, 64 and 32 bit hashes). The chains begin with the singleton configuration (all mentions in their own entity tree of size one), and at each step proposes changes to the current state which the model decides to either accept or reject. This process gradually produces larger and larger trees. For each proposed state change (sample), we record the log model ratio of both the simhash model and the exact model. We present every 10th sample in a scatter plot in Figure 1. The closer the points are to the identity reference line $y = x$, the more accurate the simhash model is for those points. Varying the number of bits has a pronounced effect on the model's ability to judge MCMC states.

For each step, we also record if the simhash model and exact model agree upon whether to accept the stochastic proposal (blue points) or not (red points).[3] The agreement rates are 97.8, 97.0, 93.6, 88.6 percent (respectively 256, 128, 64, 32 bits). We also plot the decision boundaries for this agreement as dotted lines. The upper-left and lower-right quadrants contain all the points for which the two models disagree, while the other two quadrants contain points for which they agree. In particular, the upper-right quadrants contain the points that both the simhash model and exact model believes should be accepted (true positives), while the lower-right quadrants contain the points that both models think should be rejected (true negatives). Most points lie in this quadrant since the chance of proposing a fruitful move is relatively low. Visually, there appears to be a qualitative gap between 64 bits and 128 bits on this data, leading to a recommendation of using at least 128 bits.

We also compare the models in terms of their final coreference performance (according to B-cubed F1 [Bagga and Baldwin, 1998]). The exact model achieves an F1 of 78.6, while the simhash variants achieve F1 scores of 77.6, 75.6, 62.8, 55.6 for 256, 128, 64, 32 bits respectively. For more detailed results, with precision, recall and other types of F1, see Table 1 in the appendix. Overall, the accuracy of the 128 and 256-bit models are reasonable with 256 being competitive with the performance of the exact model. When using fewer bits, again, the performance decreases precipitously.

**Experiment 2: Simhash Speed**  In this experiment we study the extent to which the compressed simhash representation improves inference speed. As described before, we tune the models on the REXA training set. Then, to test the models on a larger dataset, we supplement the 454 labeled REXA test mentions with five million unlabeled mentions from DBLP. We run each model on this combined dataset, initializing to the singleton configuration and then running one billion samples. We record coreference quality on the labeled subset every 10,000 samples and plot how it changes over time in Figures 2(a),2(b).

Although Experiment 1 shows that the simhash models are slightly worse in terms of their final F1 accuracy on the REXA test set, we see a striking computational advantage for simhash. For each F1 value, we compute how much faster the simhash model achieves that value than the exact model and plot this speed-improvement as a function of F1-level in Figures 2(c),2(d). As we can see, the speed improvement is more than a base-ten order of magnitude for most accuracy levels, and the speed difference increases with accuracy. This

---

3. These decisions, and hence agreement upon them, are deterministic with our 0-temperature sampler.

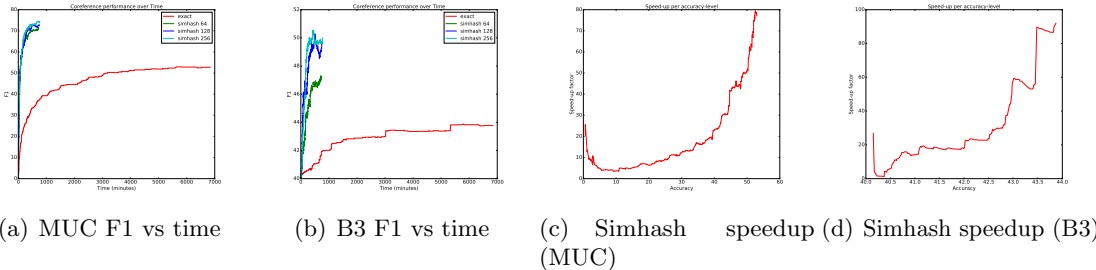

(a) MUC F1 vs time     (b) B3 F1 vs time     (c) Simhash    speedup (d) Simhash speedup (B3)
(MUC)

Figure 2: Comparison of hierarchical coreference models with either simhash or exact sparse-vector representations of the features. Simhash representations result in large speedups and have little affect on accuracy.

is because the exact representation slows down over time as the number of features in the representation grows during inference. Indeed if we look at the sampling rate over time for each model, we find that the simhash models run at about 20,000-25,000 samples per second the entire time, while the model with the exact representation starts at a rate of 3000-4000 samples per second, but then drops to under 1000 samples per second as the size of the entities get larger. This raises the question: can we improve the speed of the exact model by reducing the number of features? We address this question in Appendix C.2, but summarize here briefly: it is possible to improve the speed and reduce the gap, but simhash is still faster and there is a trade-off with coreference quality. In addition, whether feature ablation meaningfully improves performance depends on particular aspects of the data set, whereas the simhash representation can be applied generically.

**Experiment 3: JL Random Projections** In this experiment, we compare simhash with an approach that directly computes cosine on the statistics that underly the bit representation. As argued previously in Section 3.2, this approach is justified because the statistics are random projections that are homomorphic and cosine-preserving. Intuitively, we expect this approach to work even better because simhash further compresses these statistics into single bits, whereas in the random projection approach, we compute cosine directly on the real-valued statistics. However, our current theoretical analysis does not allow us to compare one to another; therefore, we turn to the empirical studies in this section.

We perform an experiment analogous to Experiment 1, except we employ the exact model to make decisions about what MCMC proposals to accept. This allows us to compare both approximation schemes on the same set of samples. For each accepted sample, we again ask the question how the approximate methods, simhash and JL, would have judged them. We find that JL does indeed perform better than simhash for the same number of bits.[4] In particuar, the Spearman's rho for JL increases from 93.2 to 97.2 over simhash when employing 256 bits (dimensions). For all but one case (128 bits), we find a similar improvement. Moreover, the coreference accuracy also increases in each case; for example,

---

4. Dimension might be a better term for JL since the floats are never converted into bits.

from 77.6 B3 F1 to 78.3. More detailed results are in Table 3 in Appendix D, along with the associated scatter plots (Figure 6).

## 5. Experiment 4: MinHash

We also investigate our homomorphic MinHash representation for Jaccard similarity. For lack of space, and because Jaccard is not employed by the hierarchical coreference, we relegate most of the evaluation to Appendix E.2. To briefly summarize, we find that 128 and 256 hash functions are sufficient for reasonable estimates and that the representation rarely needs to be refreshed to ensure that all the counts are above zero, enabling the computational efficiency we desire. However, there is room for improvement because only 16% of the hash functions on average have a non-zero entry after 100,000 samples. Thus, after 100,000 samples, a 256 hash version will begin to behave like a 40 hash version.

## 6. Related Work

Homomorphic representations have many possible uses in machine learning; for example, *homomorphic encryption* allows models to run directly on *encrypted* data for cases in which privacy is a concern [Graepel et al., 2012, Dowlin et al., 206]. Instead, our method is similar to *homomorphic compression* [McGregor, 2013] which allows our model to run directly on *compressed* data for cases in which computational efficiency is a concern. Our approach is based on simhash, a locality-sensitive hash function. Locality sensitive hashes such as simhash and `minhash` are useful in large scale streaming settings; for example, to detect duplicate webpages for web-crawlers or in nearest neighbor search [Broder, 1997b, Manku et al., 2007, Leskovec et al., 2014]. They are sometimes employed in search and machine-learning applications including coreference for "blocking" the data in order to reduce the search space [Getoor and Machanavajjhala, 2012]. Note that this application of locality sensitive hash functions for coreference is complementary to ours, and does not require it to be homomorphic. Other hashing and sketching algorithms are also employed as a strategy to compress the sufficient statistics in Bayesian models so that they can scale better as the number of parameters increase with the dataset. For example, feature hashing, count-min sketches and approximate counters have been employed in scaling latent Dirichlet categorical models such as LDA by compressing, for example, the counts that assign latent variables to mixture components [Zaheer et al., 2016, Tassarotti et al., 2018].

Our paper focuses on three homomorphic compression schemes including one for minhash. Our solution of furnishing the minhash values with counts resembles a strategy that similarly employs counts in a $k$ minimum values (KMV) sketch for estimating the number of distinct values in a set [Beyer et al., 2007, 2009]. A key difference is that that work develops an unbiased estimator for the number of *distinct values* that explicitly involves the counts as part of the estimate itself, whereas we develop a biased estimator that directly estimates *Jaccard similarity* and that employs the counts to bound our knowledge about possible minimum hash values when they vanish during difference operations.

Our schemes are related to random projections, which are commonly used to improve computational efficiency in machine learning. Examples include feature-hashing [Weinberger et al., 2009], fast matrix decomposition [Halko et al., 2009] and fast kernel computations

[Rahimi and Recht, 2007]. However, while some of these random projections happen to be homomorphic, this property is often not exploited. The reason is that they are typically used to compress static objects that remain fixed, such as the Gram matrix for kernel methods. In contrast, our setting requires compressing dynamic sets that change during inference.

Word embeddings [Mikolov et al., 2013] also provide low-dimensional dense representations that are useful for capturing context, but in practice, these embeddings are too smooth to be useful as the sole representation for disambiguating the names of peoples, places or organizations, as is necessary in coreference (e.g., the names "Brady" and "Belichick" would be highly similar according to a word-to-vec style embedding, even though they are unlikely to be coreferent). Deep learning might allow for more suitable representations to be learnt and its application to coreference is promising. However, the current research focus is on within-document noun-phrase coreference and entity linking, while emphasizing improvements in accuracy rather than inference speed and scalability [Globerson et al., 2016, Wiseman et al., 2016, Lee et al., 2017, Raiman and Raiman, 2018]. Combining deep learning and conditional random fields for hierarchical coreference remains an open problem.

Finally, in addition to the work mentioned throughout the paper, there is an abundance of related work on coreference resolution including noun-phrase coreference, cross-document coreference, record linking, entity linking and author coreference. While not possible to cover the breadth of the space here, we refer the reader to a few useful surveys and tutorials on the subject [Ng, 2010, Getoor and Machanavajjhala, 2012]. More recently, as mentioned in the foregoing paragraph, deep learning approaches are beginning to show promise. There has also been promising recent work on scalable hierarchical clustering known as PERCH [Kobren et al., 2017]. Since PERCH employs Euclidean distance rather than cosine similarity, it currently falls outside the purview of our study. However, the Johnson-Lindenstrauss applies to Euclidean distance making random projections a possibility for PERCH.

## 7. Conclusion

In this paper we presented several homomorphic compression schemes for representing the sparse, high dimensional features in a graphical model for coreference resolution, even as these features change during inference. Our primary concern was cosine similarity for which we investigated simhash and angle-preserving random projections. We also proposed a homomorphic version of minhash for Jaccard similarity. In the particular case of simhash, we presented a new variant of the original estimator and analyzed its statistical properties — including variance, bias and consistency — to help justify its use in a probabilistic model. We found that both simhash and angle-preserving random projections were sufficiently accurate, given enough dimensions, to represent features for coreference, and that the random projections produce slightly better estimates. Moreover, these representations were an order of magnitude faster than conventional sparse data structures, laying the foundation for greater scalability.

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
