# OpenReview forum: "Scaling Hierarchical Coreference with Homomorphic Compression"
_AKBC.ws/2019/Conference — AKBC 2019_

### Official Review · AnonReviewer1 · 2019-01-08
**Interesting study of simhash for a coreference resolution algorithm**

**Rating:** 6
**Confidence:** 3

**Review:**

This paper proposes to use a variant of simhash to estimate cosine similarities in a particular coreference resolution algorithm called hierarchical coreference. The original algorithm maintains many different feature sets (for mentions and groups of mentions) subject to union and difference operations, and frequently needs to estimate the cosine similarity between various such sets. To avoid the associated costs, the paper proposes to use sketching techniques instead. The proposed techniques are simple but, as the study shows, can be very effective. I like the paper overall. My main nitpick is that it's unclear whether the proposed techniques are useful for other, less specific tasks as well.

Pros:
- Simple techniques
- Analysis given
- Convincing experimental results in the considered application
- Very clear presentation

Cons:
- Quite specific, potentially little impact
- Somewhat straightforward
- Relationship to other coreference resolution methods unclear
- NEW AFTER REVISION: unclear relationship to AKMV sketch (see D3)

Detailed comments:

D1. What's a "bag type" in Sec 4?

D2. On the one hand, I like the tutorial style that the paper is partly written in. On the other hand, large parts of the (initial) discussion are not directly related to the contribution of the paper; this part could be shortened.

D3. The solution to homomorphic simhash is quite obvious. The solution to homomorphic minhash is reminiscent to the AKMV sketch of Beyer et al., "On Synopses for Distinct-Value Estimation Under Multiset Operations", SIGMOD 2007. (What's different?)

D4. Has the estimator $C_n$ been used before? If not, this might be further highlighted.

D5. It may help give a name to the estimator in 3.2. as well as to spell out its definition.

D6. I found the agree/disagree notation in Fig 1 somewhat misleading. What does it mean for the two models to agree? The decision whether to accept/reject is probabilistic.

D7. What is the total size of all sketches maintained by the algorithm in the various experiments? (It appears 1kb per node [my rough guess] is quite substantial, although it may be less than in the exact method, at least for some nodes.)

D8. It would also be interesting to see the performance w.r.t. number of steps taken.

D9. Is it possible to speed up the exact method to obtain similar performance improvements? Has the method been tuned (e.g., to that fact that most proposals appear to be rejected?)

D10. It remains unclear how the proposed hierarchical coreference model relates to state-of-the-art models, both in terms of cost and in terms of performance. This is a weak point: one may have the impression that this paper speeds up an method that is not state-of-the-art anymore.

Typos:

"We estimate parameters with hyper-parameter search"

---

### Official Review · AnonReviewer3 · 2019-01-17
**Interesting paper, incomplete experiments**

**Rating:** 5
**Confidence:** 3

**Review:**

This paper proposes to use homomorphic compression for hierarchical coreference resolution. Contributions of the paper are threefold. First, it proposes a homomorphic cosine preserving hashing version of simhash. Secondly, it presents another homomorphic cosine preserving representation based on random projections. Thirdly, the paper proposes a homomorphic version of minhash for Jaccard similarity. The paper applies these representations to the hierarchical coreference resolution problem using CRF. The authors also provide statistical analysis. Experimental results on real-world datasets are also presented. The paper is generally well written.

My main concern with the paper is that it falls short of comparing with other valid baselines. One of the main points of the papers is that sparse and high-dimensional representations of mentions creates problems during probabilistic inference. In order to overcome this problem, the authors propose various hashing-based methods. However, in light of recent advances in word embeddings, one is not limited to using such sparse and high-dimensional representations. One can potentially use off-the-shelf dense and relatively lower-dimensional embeddings such as word2vec, glove etc, or use context-dependent embeddings such as ELMO. Unfortunately, the paper completely ignores this line of research, and neither compares nor cites them. This is clearly not satisfactory, and the paper is not complete with such vital missing pieces.

Overall, I think the paper proposes some interesting ideas, but it is incomplete in light of the issues above. Moreover, applying the proposed hashed representation on at least one other task will make the paper stronger.

---

### Official Review · AnonReviewer4 · 2019-01-24

**Rating:** 7
**Confidence:** 3

**Review:**

-summary
This paper addresses the problem of model scaling and efficiency in coreference resolution. Commonly used models represent mentions as sparse feature vectors which are iteratively agglomerated to form clusters of mentions. The feature vector of a cluster is a function of all the mentions in the cluster which causes them to become more dense over time and computationally expensive to operate over. To address this, the authors investigate hashing schemes to expedite similarity calculation. Importantly, they propose a homomorphic hash function that is able to update the representation of a cluster online as new mentions are added or removed.


-pros
 — a homomorphic simhash that can speed up similarity calculations while maintaining performance
 — experiments  show that the hashing schema is able to perform within 1 F1 point of the exact similarity model.
 — good coverage of related work and explanation of methods

-cons
— only tested on a single dataset
— the lines in Fig 2 are not the clearest presentation of the speed accuracy tradeoff due to the large differences in scales and the fact that the maximum accuracies are never reached for some lines. Maybe a table or another presentation of these results could help.

---

### Meta-Review · Area_Chair1 · 2019-02-12
**Weak accept**

**Recommendation:** Accept (Poster)
**Confidence:** 4

**Metareview:**

This paper proposes to use homomorphic compression for hierarchical coreference resolution. The proposed method is novel and experimental results show good results. The paper is well written.

However, there are crucial omissions that the authors should have addressed in a revision, such as a detailed comparison w. [Beyer et al.], empirical comparison w. a Word2Vec embedding baseline, etc. We very strongly encourage the authors to include them in the final version.

---

### Decision · Program_Chairs · 2019-02-15
**AKBC 2019 Conference Decision**

Accept